# Serum iron, Magnesium, Copper, and Manganese Levels in Alcoholism: A Systematic Review

**DOI:** 10.3390/molecules24071361

**Published:** 2019-04-07

**Authors:** Cezary Grochowski, Eliza Blicharska, Jacek Baj, Aleksandra Mierzwińska, Karolina Brzozowska, Alicja Forma, Ryszard Maciejewski

**Affiliations:** 1Chair and Department of Anatomy, Medical University of Lublin, 20-090 Lublin, Poland; jacek.baj@me.com (J.B.); maciejewski.r@gmail.com (R.M.); 2Department of Neurosurgery and Pediatric Neurosurgery, Medical University of Lublin, 20-090 Lublin, Poland; 3Department of Analitical Chemistry, Medical University of Lublin, 20-090 Lublin, Poland; bayrena@o2.pl; 4Department of Forensic Medicine, Medical University of Lublin, 20-090 Lublin, Poland; mierzwinska.aa@gmail.com (A.M.); brzozowskak@gmail.com (K.B.); aforma@o2pl (A.F.)

**Keywords:** alcoholism, alcoholic liver disease, iron, magnesium, copper, manganese, deficiency

## Abstract

The aim of this paper was to review recent literature (from 2000 onwards) and summarize the newest findings on fluctuations in the concentration of some essential macro- and microelements in those patients with a history of chronic alcohol abuse. The focus was mainly on four elements which the authors found of particular interest: Iron, magnesium, copper, and manganese. After independently reviewing over 50 articles, the results were consistent with regard to iron and magnesium. On the other hand, data were limited, and in some cases contradictory, as far as copper and manganese were concerned. Iron overload and magnesium deficiency are two common results of an excessive and prolonged consumption of alcohol. An increase in the levels of iron can be seen both in the serum and within the cells, hepatocytes in particular. This is due to a number of factors: Increased ferritin levels, lower hepcidin levels, as well as some fluctuations in the concentration of the TfR receptor for transferrin, among others. Hypomagnesemia is universally observed among those suffering from alcoholism. Again, the causes for this are numerous and include malnutrition, drug abuse, respiratory alkalosis, and gastrointestinal problems, apart from the direct influence of excessive alcohol intake. Unfortunately, studies regarding the levels of both copper and manganese in the case of (alcoholic) liver disease are scarce and often contradictory. Still, the authors have attempted to summarize and give a thorough insight into the literature available, bearing in mind the difficulties involved in the studies. Frequent comorbidities and mutual relationships between the elements in question are just some of the complications in the study of this topic.

## 1. Introduction

Fluctuations in the concentration of minerals, vitamins, and ions in the human body are common among those patients who consume excessively high amounts of alcohol. Changes in these amounts are observed mainly due to an inappropriate nutritional status, alcohol-induced vomiting, diarrhea or excessive urination. Deficiency or excessive amounts of particular ions may induce more or less severe dysfunctions, including impairments in the proper functioning of the cardiovascular, nervous, and skeletal systems. In the following review paper, mainly iron, magnesium, manganese, and copper levels among alcohol-dependent people have been investigated. The main aim was to investigate potential changes in the concentration of these ions under the condition of chronic increased alcohol consumption. Over fifty articles have been reviewed by four independent authors, dating from the year 2000 onwards. The results were mostly conclusive with regard to the levels of iron and magnesium in those patients who suffer from alcohol dependence. All reviewed papers indicate an increase in the levels of serum iron and hepatic iron overload [1]. Iron is an essential element in a number of biochemical reactions. It forms complexes with oxygen, mainly in hemoglobin and myoglobin. Iron is also found in the active sites of enzymes responsible for oxidation and reduction reactions. The accumulation of iron in the liver is associated with increased ferritin synthesis and a reduced amount of hepcidin produced. Due to an increased synthesis of ferritin, more iron can be stored in the body [2]. Additionally, pathologically decreased amounts of hepcidin lead to an accumulation of iron. This is because of an increased efflux of these ions to the outside of the cell via ferroportin, thus increasing the level of iron in the bloodstream. Magnesium is an essential element responsible for the proper function of a number of enzymes in the organism [3]. Magnesium ions, for instance, are needed for the enzymes active in the production of ATP. Additionally, an ATP molecule itself is found in the form of a chelate with magnesium ion. Copper is an essential transition metal that acts as a cofactor for a number of enzymes, thus ensuring proper metabolism and homeostasis. It participates in the metabolism of lipids, as well as redox balance and iron mobilization—with the latter being an interesting relationship to be further explored in the context of (alcoholic) liver disease and iron overload. Finally, the role of manganese in the body’s metabolism is mostly thanks to its function as a coenzyme: It is a structural part of arginase (essential in the proper metabolism of urea), among others. Manganese also acts as an activator of numerous enzymes in the Krebs cycle, particularly in the decarboxylation process. It is essential for proper bone growth. In higher concentrations, manganese can act as a neurotoxin which is then deposited in the brain, in particular the basal ganglia, affecting proper function [4]. Chronic alcohol consumption leads to structural changes in the brain [5,6,7]. Unfortunately, studies regarding the levels of both copper and manganese in the case of (alcoholic) liver disease are scarce and often contradictory. Still, the authors have attempted to summarize and draw conclusions, as well as propose potential future directions for research from the literature currently available.

### Healthy Ranges

Iron levels differ between individuals and can indicate a number of metabolic disorders [8]. A healthy adult should have about 1700–2000 mg of iron in the form of heme iron, about 130–150 mg as a cofactor of a number of enzymes, as well as up to 1000 mg of stored iron held in reserves. An increase in iron stores is commonly seen in hemochromatosis as well as in non-alcoholic fatty liver disease (NAFLD), as a manifestation of the metabolic syndrome [9]. On the other hand, iron stores deficiency is common among patients suffering from anemia [10]. Ferritin is the most sensitive and specific indicator of iron deficiency [11,12]. Ferritin levels below 30 μg/L are an indicator of iron deficiency. Other serum markers of iron stores include transferrin–iron saturation [TS] and serum iron levels themselves [13,14,15]. All of the abovementioned markers are higher in people who consume mild or moderate amounts of alcohol compared to those in nondrinkers [16].

The total magnesium body content of a healthy individual should oscillate at around 20 mmol/kg of fat-free tissue. That is, an average healthy adult individual weighing 70 kg will have around 25 g of total body magnesium. Ninety-nine percent of the body’s magnesium is in the form of intracellular magnesium and hence, magnesium serum levels cannot be used to diagnose magnesium deficiency or overload. 

In physiological conditions, a healthy adult body contains 80 mg of copper with the concentration being highest in the eye, the heart, the liver, and the brain. The serum copper levels should fall at around 109 μg/100 mL, with 90% bound in the form of ceruloplasmin and the remaining copper loosely associated with serum albumins. Increased copper levels are associated with Alzheimer’s disease.

Healthy ranges of manganese levels are 4–15 μg/L in the blood, 1–8 μg/L in urine, and 0.4–0.85 μg/L in serum. Excess manganese tends to accumulate in the basal ganglia of the brain. Normal manganese levels in the brain average 1–2 µg/g dry weight and vary depending on the region of the brain. An increase in the brain’s manganese levels is associated with neurological disorders, Parkinson’s disease in particular.

## 2. Iron Levels in Alcoholism

### 2.1. Concentration of Iron in the Liver

Pathologically high amounts of ferritin are usually diagnosed accidentally through screening tests or typical check-ups. The most common causes for high concentration of iron in the organism are alcoholism, inflammation, cytolysis, and metabolic syndrome. Patients with chronic hematologic diseases (either acquired or congenital), or those whose iron supplementation is excessively high, like alkalized patients or sportsmen, are particularly exposed to the risk of hyperphosphatemia. If ferritin concentration is excessively high, congenital hemochromatosis must initially be taken into account and excluded as the causative factor. Magnetic resonance imaging (MRI) of the liver is a useful tool in the diagnosis and treatment of hemochromatosis [17]. The majority of the serum ferritin is in a glycosylated form (60–90%), and macrophages are the source of it. The nonglycosylated fraction constitutes approximately 20–40%, with cytolysis being the major source. Appropriate levels of ferritin in the organism are 30–300 mg/L and 15–200 mg/L for men and women, respectively. Hyperphosphatemia occurs frequently in the case of chronic alcoholism. The prevalence varies from 40% even up to 70%. Interestingly, iron levels in the organism are not proportional to the amount of consumed alcohol. This can be explained because of the direct effects of alcohol. Ethanol induces synthesis of ferritin and decreases the amount of produced hepcidin in the body. Normally, ferritin concentration remains at <1000 mg/L, and the concentration of transferrin remains at a regular level as well. Among alcoholics, the concentration of ferritin exceeds 1000 mg/L, and the saturation of transferrin exceeds 60%. Excessively high levels of both transferrin and ferritin can be observed for up to 6 weeks after alcohol withdrawal. Additionally, other clinical consequences associated with excessive alcohol consumption, including acute or chronic hepatitis, might cause an increase in the levels of ferritin even up to >10.000 mg/L. During inflammation, a number of proinflammatory cytokines are released, including IL-1, IL-6, and TNF-alfa [18]. Particularly IL-6 stimulates the synthesis of ferritin and hepcidin. Increased amounts of hepcidin stimulate the uptake of iron by the enterocytes and macrophages, which is associated with increased synthesis of ferritin [19].

### 2.2. Iron Accumulation and the Role of Hepcidin

An excessive consumption of alcohol is responsible for a disturbance in iron metabolism in the organism. Accumulation of iron can damage a number of different organs—mainly the liver. Moreover, deficiency of iron due to macrocytic anemia or accumulation of iron in the liver among people who consume excessive amounts of alcohol can lead to hematologic disorders. One of the functions of hepatocytes is the synthesis of hepcidin, which is associated with control of iron accumulation and the location of iron in the organism. Hepcidin inhibits the transport of iron from enterocytes to the blood. It was observed that increased alcohol consumption lowers the expression of the gene coding hepcidin. Moreover, alcohol is proven to cause intracellular oxidative stress [20]. This is enhanced due to iron accumulation in the liver leading to hepatic steatosis, fibrosis, and eventually cirrhosis.

### 2.3. sTfR Receptor as a Marker in Alcoholism

Distortions of the iron metabolism among alcoholics are associated with deficiency or excessive amounts of iron in the organism, which may lead to hematologic disorders. Transferrin receptor TfR is a glycoprotein localized in the cell membrane of immature erythrocytes, which are found in the bone marrow. Its function is the transport of iron to the inside of the cells, previously binding to transferrin. Synthesis of the TfR receptor and transferrin depends on the activity of erythrocytes and the need for iron in the organism. Recently, sTfR was observed to be a beneficial marker in the diagnosis of iron deficiency, while other markers like hemoglobin levels or mean corpuscular volume (MCV) are at an appropriate level. Research has shown that average percentages of carbohydrate-deficient transferrin (CTD), alanine aminotransferase (ALT), gamma-glutamylotransferase (GGT), and MCV were significantly increased in the group of alcoholic patients in comparison to the control group. It was also observed that transferrin levels were not changed between the studied groups. However, transferrin iron saturation was much higher in the group of alcohol-dependent people. The level of sTfR is a beneficial method for the identification of iron deficiency in the organism. However, some conditions must first be followed, which are that other markers remain undisturbed and that sTfR is not strictly associated with acute phase reactions. There is a correlation between sTfR level and erythropoiesis and iron demand in the organism. Among healthy people, approximately 80% of this receptor is found in the bone marrow. The level of sTfR is therefore strictly associated with the activity of erythropoiesis in the bone marrow. This means that in case of anemia due to iron deficiency, the level of this receptor increases in the serum. Nevertheless, information is lacking when it comes to sTfR concentration in the conditions of excessive iron concentration in the body. This explains why the normal concentration of sTfR among alcoholic patients may inform about proliferation in the course of erythropoiesis. Since this level remains unchanged, iron absorption through transferrin receptor remains intact as well. A likely explanation for this is that the sTfR level depends on the suppression of hepcidin expression induced by ethanol. In the case of excessive absorption of iron in the intestinal tract (and thus, higher iron concentration in the blood), the ability of iron transport to the enterocytes and the speed of erythropoiesis remain intact. The concentration of sTfR in the serum does not depend on the metabolic status of the hepatocytes, age, and duration of alcohol addiction or alcohol withdrawal. As such, it can be concluded that neither alcohol consumption nor alcohol abstinence change the amount of transferrin receptors found in the serum.

### 2.4. Concentration of Iron in the Brain

Chronic alcohol abuse can lead to abnormally high amounts of iron in the organism, including in the brain [9]. An overload of iron can lead to neurotoxicity in the brain [21]. However, it is important to note that iron concentration is normally rather high due to its high metabolic rate and the need for iron as an enzymatic cofactor for myelination or catecholamine synthesis. 

One of the methods that enable the monitoring of iron concentration in the brain is an MRI technique called quantitative susceptibility mapping [22,23]. The results of this research paper have shown an increase of iron concentration in all regions of the brain, with some differences in the deposited amounts depending on the specific region in the brain. The increase in iron concentration varied from 7% to 15%. Most significant differences were observed mainly in the caudate nucleus, the putamen, and the globus pallidus, as well as in the dentate nucleus. Such high concentrations of iron in deep grey matter are comparable to those in the liver in some cases [24].

Some evidence suggests that brain function restoration is possible following a period of abstinence from alcohol [25].

## 3. Magnesium Levels in Alcoholism

### 3.1. Alcohol as a Reason of Hypomagnesemia

Hypomagnesemia is a frequently occurring electrolyte disorder due to a number of possible causes. These include drug abuse, malnutrition, respiratory alkalosis, excessive alcohol intake or gastrointestinal problems. According to a recent report published by the Mayo Clinic, the risk of developing hypomagnesemia among people who excessively consume alcohol varies from 30% even up to 80% [26]. This condition is observed when the serum magnesium concentration is below 0.66 mmol/L. However, clinical symptoms are observed when it falls further still, below 0.5 mmol/L.

Severe deficiency of magnesium ions can lead to a severe imbalance in the body’s homeostasis [27,28]. Magnesium is essential for protein synthesis and a number of enzymatic reactions [29]. Furthermore, there is a correlation between the amount of magnesium ions and a number of other elements, including phosphorus, calcium or potassium [30]. As such, any imbalance in one of these may eventually induce other consequences similar to those caused by magnesium deficiency. Magnesium deficiency, for instance, leads to hypocalcemia, since it affects the magnesium-dependent adenyl cyclase generation of cyclic adenosine monophosphate. As a consequence, this decreases the amount of parathormone in the body. Therefore, it leads to changes in the functioning of the cell membranes and causes disorders of the nervous and cardiac systems, as well as inappropriate functioning of the muscles [31]. The pathogenic mechanisms of hypomagnesemia also include magnesuria, which may be related to hypophosphatemia, metabolic acidosis, or direct effect of the alcohol on the amount of the ions in the organism [32].

### 3.2. Concentration of Ionized Magnesium in the Erythrocytes and Blood Plasma

A disturbed pattern of magnesium levels in the organism is commonly observed among those patients with a history of chronic alcohol abuse. Ionized magnesium constitutes approximately 67% of the total pool of the magnesium in the body, but only 1% of this amount is found in the blood serum. As such, the relationship between extracellular and intracellular ion concentration can be studied, which may present more clearly the effect of the intake of the alcohol on these amounts. Hypomagnesemia of alcohol-dependent patients is mainly due to alcohol-induced vomiting and diarrhea, excessive urination, and an inappropriate diet lacking in a proper vitamin and microelements supplementation. The research so far is consistent in reporting that alcohol consumption only affects the amount of the ionized magnesium, with little or no effect on the total magnesium concentration [33].

In the study performed by Ordak et al. [34], the amount of ionized magnesium in the erythrocytes and blood plasma of both alcohol-dependent people and the control group was investigated. The results indicated that the amount of magnesium in the erythrocytes in the alcohol-dependent group was almost twice as low compared to that in the control group. However, the amount of ionized magnesium in the blood plasma in both groups remained unchanged. Interestingly, the total amount of magnesium in the blood plasma and the erythrocytes differed between the two groups. A significant decrease was observed in the alcohol-dependent group compared to the control group. Additionally, SF-36 questionnaires performed in both groups showed that individuals with lower total magnesium concentration tend to have a lower quality of life. They were more impulsive and susceptible to mental disturbances, as well as suffered from sleeping disorders. Furthermore, a correlation was found between the amount of ionized magnesium and physiological changes as analyzed by the Bried symptom inventory, the Barratt impulsiveness scale, and the sleep disorder questionnaire.

### 3.3. Hypomagnesemia and Cardiovascular System

Alcohol consumption may also lead to changes of the cardiac conduction system. The cardiovascular system is usually affected when ethanol consumption exceeds 80g daily for approximately 10 years. However, lower daily doses for a longer period of time or greater doses for shorter than ten years may also lead to similar clinical consequences [35,36].

One of the dysfunctions associated with this observation may be prolongation of the QT interval. In the study performed by Moulin et al. (2015) [37], groups of both active and abstinent alcoholics were studied. The results showed that active alcoholics exhibited a higher heart rate. Furthermore, 12 out of 166 individuals, among whom 10 were active alcoholics, were observed with prolonged QT interval. Hypomagnesemia has been shown in 50% of those individuals with a prolonged QT interval. The study showed that active alcoholics were three times more likely to develop hypomagnesemia, and at least nine times more likely to exhibit a prolonged QT interval. This suggests that hypomagnesemia caused by alcoholism may be a relevant factor in the prolongation of the QT interval. In a case study performed by Hiroki Nakasone et al. (2001), a patient suffering from alcoholic liver cirrhosis was observed to have critical arrhythmia (torsade de pointes, TdP). The laboratory data showed hypomagnesemia, which could potentially lead to the development of TdP. Therapy included a supplementation of magnesium sulfate to restore appropriate magnesium levels. It can be concluded that patients with an alcoholic liver disease, along with hypomagnesemia and hypocalcemia and a prolonged QT interval, are at higher risk of developing torsade de pointes.

Other studies, including research by Paulo Borini et al. (2001), presented similar results of prolonged QT intervals in female alcoholics alike [38]. 

### 3.4. Effects of the Hypomagnesemia on the Digestive System

Considerable magnesium deficiency has also been observed among alcoholics with pancreatitis caused by excessive alcohol intake. Magnesium deficiency was higher among alcoholics with pancreatitis than among those where this condition was not present [39,40]. The research concludes that continuous intake of alcohol may lead to the greater magnesium deficiency over time.

In the study performed by Turecky in 2006, a group of 44 patients with liver steatosis (either alcoholic or non-alcoholic) and a control group were investigated [41]. The study explored whether alcohol consumption is responsible for magnesium deficiency and whether liver disorders may also play a significant role. The results showed hypomagnesemia in both groups. Patients suffering from hepatic steatosis showed distortions in the secretion of bile acids as well. This consequently lowers the amount of endogenous magnesium concentration, as the increased amounts of fatty acids in the intestinal lumen form insoluble soaps with magnesium.

Since hypomagnesemia was also observed in the group of non-alcoholic patients, it must be considered that alcoholism is not the only cause of hypomagnesemia among patients with hepatic steatosis. 

### 3.5. Alcohol Intake and Paralysis

One severe consequence of excessive alcohol intake is paralysis associated with both hypokalemia and hypomagnesemia. The continual excretion of minerals and vitamins from the body may induce muscle weakness due to inappropriate amounts of ions, mainly in the intracellular environment. The treatment in the form of infusion of thiamine, glucose, potassium, and magnesium results in quick improvement of hypomagnesemia [42]. On the other hand, hypokalemia might not be improved that quickly. Nevertheless, muscle performance appears to be improved, and symptoms of the paralysis are significantly lowered [32]. Since hypomagnesemia-induced kaliuresis eventually leads to hypokalemia, potassium replacement may play a role in the treatment of paralysis.

### 3.6. Hypomagnesemia and Neural Dysfunctions

Since hypomagnesemia is a common electrolyte disorder associated with excessive alcohol intake, it has a great impact on every system of the organism. It includes more or less severe implications on the nervous system. One of such dysfunctions may be childhood seizures with non-neurological etiology but as a cause of hypomagnesemia [43]. In all of the cases concerning childhood seizures with no neurological background, a decreased level of magnesium has been observed. Additionally, when magnesium supplementation has been provided, the intensity of seizures was lowered. The exact mechanism of decline of magnesium and seizures has not been proposed. Nevertheless, it was studied on a rat model which showed that a decreased level of extracellular magnesium leads to lack of antagonism at the N-methyl-D-aspartate-type glutamate receptors, thus resulting in epileptiform discharges.

Even though it may be not that common, hypomagnesemia can lead to such implications as posterior reversible encephalopathy syndrome (PRES). This condition is also induced by various factors like high blood pressure, autoimmune disorders, renal failure or immunosuppressive therapy. However, there are cases of PRES induced by hypomagnesemia noted in the literature. This was confirmed by the chemical analysis of serum magnesium levels in patients with PRES. In this case, the parenteral replacement of magnesium induced rapid clinical improvement. 

### 3.7. Hypomagnesemia and Level of Other Ions

A two-year-long hospital study was performed in 2012 by George Liamis et al. [44]^.^ Patients chosen for the following study were a group of 107 nonselected, consecutive, adult patients (over 18 years of age) who were observed with hypomagnesemia. Among the examined group, 13 patients (12.1%) developed hypomagnesemia due to the excessive alcohol consumption. The study also showed a link between magnesium deficiency and other elements, including potassium and calcium. Additionally, the study indicated a higher incidence of hypomagnesemia in patients over the age of 65. Thus, it should be considered that hypomagnesemia induced by alcohol consumption may be more or less severe when other factors are taken into consideration. This includes the presence of other disorders like diabetes mellitus or pancreatitis, the intake of medicines such as loop and thiazide diuretics or proton pump inhibitors, age or nutritional status.

## 4. Copper levels in Alcoholism

### 4.1. Copper in the Case of Alcoholism: Deficiency or Increased Levels?

While there are a number of consistent reports on selenium and zinc deficiency and excessive alcohol consumption, reports on copper are scarcer and often inconclusive, or indeed lead to contradictory results [45]. On the one hand, copper deficiency has been reported in a number of studies on alcoholic liver disease (ALD), as well as malnutrition linked to alcoholism [46,47]. However, some studies (Rahelic et al., 2006) have also reported an increase in copper levels of patients suffering from alcoholism, as linked to zinc deficiency [48]. This goes to show just how intricate the relationship between copper levels and other metabolic conditions is [49]. Since the liver is a central organ for copper metabolism, copper deficiency has been reported in diseases where the metabolism of lipids has been disrupted, including non-alcoholic liver disease (Eigner at al., 2008), as well as liver cirrhosis in the course of alcoholism [50]. Other common diseases linked to insufficient copper concentration are obesity, metabolic syndrome, hematological disorders, and ischemic heart disease [51,52]. However, none of the above relationships have been extensively studied, and reports from 2000 onwards are lacking. 

### 4.2. The Relationship Between Copper and Zinc

Serum copper levels remain in a close relationship with the levels of zinc in an inversely proportional relationship. The mechanism of this interaction is yet to be fully understood, but it is certain that high zinc consumption inhibits copper absorption. A difficulty in reporting and comparing copper levels is the fact that copper is mostly an extracellular element (90%), while the contents of zinc in blood serum are low at 1%, and zinc is mostly found in the intracellular environment. This has been accounted for in the study by Ordak et al., 2017, where the relationship between the concentrations of zinc in erythrocytes/copper in blood plasma in alcohol-dependent patients and the clinical parameters was explored [53]. It was concluded that copper deficiency in alcohol-dependent patients often correlates to reduced and impaired function of the central nervous system, thus affecting patients’ mental and physical state. Indeed, since any increase in zinc absorption can lead to a significant copper deficiency, maintenance of the proper copper to zinc ratios is essential to the body’s homeostasis. Both the levels of zinc and copper are affected in the case of alcoholism, consequently impairing day-to-day functioning. 

An imbalance in the levels of copper and zinc has been linked to Alzheimer’s disease (AD) (Mital et al., 2018; Sensi et al., 2018) [54,55]. Pathogenesis of Alzheimer’s disease involves accumulation of the β-amyloid (Aβ) peptide in the brain. The activity of the Zn^2+^-dependent endopeptidase neprilysin (NEP) remains in an inverse relationship with brain Aβ levels during aging and in AD—as such, zinc acts as a protective agent. An increase in the levels of copper, on the other hand, will have a negative effect both through the modulation of zinc concentration, as well as directly, since copper ions inhibit the proper function of NEP. With regard to cognitive function, copper is also linked to certain psychiatric disorders, including depression [56]. Moreover, copper deficiency can be a cause of idiopathic myelopathy in adults [57,58,59].

### 4.3. The Relationship Between Copper and Iron

Copper availability contributes to iron metabolism and homeostasis, as described by Eigner et al. 2008 [50]. However, the relationship between copper serum and liver content was explored here only in the context of patients suffering from non-alcoholic fatty liver disease (NAFLD). No similar studies have been performed with regard to hepatic iron concentrations and alcoholic liver disease (ALD). Still, it may be worthy to note that the low bioavailability of copper observed in NAFLD causes increased hepatic iron stores. This is due to a decrease in the expression of ferroportin-1 as well as the activity of ceruloplasmin ferroxidase, both of which lead to a block in the export of iron from the liver. As both copper deficiency and iron overload are commonly observed in patients with ALD, a corresponding relationship may be worth exploring in future studies. The same study also reported low liver copper concentrations in the context of increased insulin resistance and metabolic syndrome. 

### 4.4. Copper Deficiency and Lipid Metabolism

Another notice of copper insufficiency linked to lipid synthesis and fatty liver disease was made in the paper by Morrell et al., 2017 [51]. Inadequate levels of copper may further promote the damaging effects of excessive alcohol consumption. Copper deficiency promotes dyslipidemia and increases oxidative stress, since copper is an essential cofactor of a number of antioxidant enzymes. Similarly, the study by Ordak et al. found decreased serum copper concentration among 20.4% of the examined patients with alcohol abuse history [53]. This also corresponded to worsening of the symptoms related to the central nervous system, including depressive symptoms and sleep deficiency.

### 4.5. Serum Copper Levels and Hepatitis C Virus (HCV) Infection 

In addition to alcohol consumption, the relationship between serum copper levels (as well as zinc and selenium) and HCV infection was investigated in a paper by Gonzalez-Reimers et al. (2009) [60]. In this research paper, no interaction was found between serum copper levels and alcohol consumption, but a direct relationship was confirmed between serum Cu levels and HCV infection. Serum copper levels were significantly lower in those patients who tested positive for HCV. Moreover, the relationship between copper and serum malonaldehyde (MDA), as well as a number of cytokines, was explored, but no conclusive relationship was confirmed.

### 4.6. Copper Muscle Content 

Finally, an interesting study by Duran Castellon et al. (2005) looked at muscle content of several elements in order to explore the pathogenesis of muscle myopathy as a common occurrence in patients with alcoholism [61]. This seems to be a combined effect of alcoholism and protein malnutrition. Increased protein catabolism, as well as protein deficiency stemming from malnutrition is commonly seen in alcoholism and may be linked to copper depletion. This leads to muscle myopathy. However, in this study, ethanol was found to have no effect on muscle copper. Rather, the decreased copper content in the muscles was due to malnutrition and, in particular, lack of protein in the diet.

## 5. Manganese in Alcoholism

### 5.1. Manganese in the Case of Alcoholism: Always Overload?

Similarly to the case of copper discussed in the above paragraph, the role of manganese in the pathogenesis of liver cirrhosis and other complications in the course of alcoholism is not yet fully understood. However, studies seem to be consistent in reporting elevated levels of manganese in the course of alcoholic liver disease. In particular, the role of manganese as a neurotoxin plays a significant role in the course of development of brain disorders related to alcohol abuse, including hepatic encephalopathy and acquired hepatocerebral degeneration [48]. In contrast to the above findings, a study was performed by Gonzalez-Perez et al. (2011), which explored the relationship between ethanol consumption and manganese levels in the bones [62]. Significant decrease of bone manganese levels was found in those patients with a history of alcohol abuse.

### 5.2. Manganese and Liver Cirrhosis

The study by Rahelić et al. (2006) explored the levels of a number or trace elements in patients with liver cirrhosis, including manganese [48]. No significant difference was found in the levels of manganese between the control group and the male and female patients with cirrhosis. However, there was a significant increase in the levels of magnesium in the case of patients suffering from Child-Pugh type C liver cirrhosis compared to patients with Child-Pugh A and B liver cirrhosis. The same study further investigated the relationship between manganese levels in cirrhotic patients with or without encephalopathy and ascites. No differences were found in the case of encephalopathy, which is contradictory to some studies performed in the past, which indeed found increased manganese levels as a relation to hepatic encephalopathy. Significantly higher levels of manganese were reported in the serum of those cirrhotic patients simultaneously suffering from ascites. The study goes on to discuss manganese bile secretion and its disruption in the course of cholestatic liver disease as one possible cause for increased manganese levels. Still, it is apparent that the levels of manganese as explored simply in a relationship to liver cirrhosis can vary significantly between different research studies. 

### 5.3. Manganese as a Neurotoxin 

Studies are consistent in reporting an increased deposition of manganese in the brain of patients suffering from hepatic encephalopathy (HE) [63,64,65,66,67,68,69]. This is depicted in the form of hyperintense MRI signals, in particular in the region of the brain called the globus pallidus [70]. Some studies have reported manganese levels that were up to seven times higher in patients with alcoholic liver disease as shown on the MRI. Again, the previously mentioned bile secretion of manganese plays a significant role in manganese crossing the blood–brain barrier in those patients suffering from cholestasis. Compromised liver function does not allow for proper detoxification, and one of the results is a free passage of neurotoxins entering the brain. As increased amounts of manganese are deposited in the brain, and in particular in the astrocytes, this causes Alzheimer type II changes, as well as selective neuronal loss in the basal ganglia [66,67,68]. Similar effects are seen in ammonia deposition in the brain [69]. Apart from the globus pallidus, substantia nigra reticulata is another region particularly affected [63]. Both the structure and the proper function of the brain are affected.

Another brain disorder linked to alcohol abuse and increased magnesium concentration is acquired hepatocerebral degeneration (AHE). This condition needs to be made distinct from hepatic encephalopathy, and it affects approximately 1% of patients suffering from liver cirrhosis. It exhibits itself in the form of symptoms similar to those of Wilson disease, but without the increased copper deposition. Instead, high intracerebral manganese concentration can be seen in T1-weighted MRI images, in particular in the basal ganglia, as well as some other brain areas (pituitary gland, quadrigeminal plate, caudate nucleus, subthalamic region, and red nucleus) [70].

Finally, cirrhosis-related parkinsonism commonly affects patients with a history of chronic alcohol abuse. Again, this is due to manganese deposition in the brain and is characterized by the usual symptoms of Parkinson’s disease but without affecting the nigrostriatal system [71,72]. Instead, elevated levels of manganese can be found in the serum, as well as depicted in an MRI scan. 

### 5.4. Bone Manganese Levels and Alcohol Abuse

An interesting relationship was explored in the study by Gonzalez-Perez et al. (2011) [62]. This study reported significantly lower levels of manganese in the bone tissue of patients suffering from alcoholic disease. Since appropriate levels of manganese are required for normal bone growth, as it is a cofactor of a number of essential enzymes, both the bone mass and bone synthesis were affected. This, together with malnutrition and the deficiency of other essential elements, can be linked to the loss of bone mass and osteoporosis in patients with a history of alcohol abuse. 

## 6. Conclusions

Chronic alcohol consumption leads to a number of metabolic disorders associated with deficiency or overloading of elements in the human body. The article is a review paper concerning, inter alia, correlation of Fe, Mg, Cu, and Mn content in people who drink alcohol. The accumulation of iron in the liver is associated with increased ferritin synthesis and a reduced amount of hepcidin produced. This results in alcohol damage of the liver, which can consequently lead to liver cirrhosis. 

Excessive alcohol consumption leads to hypomagnesemia in almost all cases, differing in the severity of the deficiency. It is mainly due to malnutrition, excessive urination, diarrhea, and alcohol-induced vomiting. Interestingly, according to the research mentioned in this work, it is mainly ionized magnesium concentration which is changed. Thence, it has a significant impact on many functions and systems of the organism, including the nervous or cardiovascular system. Nicotine-dependent patients with alcoholism are at a particularly higher risk of cardiovascular disease [73]. The appropriate treatment of such a condition, like infusion of magnesium, glucose, and potassium, may give quick improvement for the patients, also lowering the harmful effects of hypomagnesemia on the organism. However, it also must be taken into consideration that deficiency of magnesium ions is strictly associated with changed concentrations of other ions like calcium, iron, manganese or copper. Therefore, there is a relationship between these values and overall effects on the organism. Moreover, other factors like age, additional diseases, and usage of medicine may have an impact on the overall concentration of the magnesium among alcohol-dependent people. 

With regard to copper levels in patients suffering from alcoholism, the few studies that have been performed so far are inconsistent. Several authors investigated serum copper levels in patients both in the case of alcoholism, as well as during periods of withdrawal and after the relief of withdrawal symptoms, and observed copper deficiency. Still, a number of similar studies reported no difference in the levels of copper between alcohol-dependent individuals and control groups. In fact, higher levels of copper were reported in the case of the former in some research studies. Certainly, the close relationship between copper and zinc, and indeed the intricate relations between the metabolic pathways of various other elements add to the complexity of research on the topic. 

The few studies which have been found by the authors seem to be consistent in reporting increased magnesium levels in the patients characterized by chronic alcohol abuse. In particular, the role of manganese as a neurotoxin is discussed. Alcohol abuse induces brain atrophy and neuronal loss, as well as myelin sheath degeneration, all of which seem to be reversible to some extent following periods of alcohol abstinence [74,75]. One study explored the reduction in the levels of manganese in the bones as a result of alcohol abuse.

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
