# Peer review of "Serum iron, Magnesium, Copper, and Manganese Levels in Alcoholism: A Systematic Review"

_molecules, 2019, doi:10.3390/molecules24071361_

Round 1

Reviewer 1 Report

In this manuscript, the authors provide a good overview of the current state of knowledge about changes in copper, manganese, iron and magnesium in alcoholic patients. The review is properly written and can be followed easily. 

Comments: 

Section 2.4; lines 158, 169, 170: In the title of this section, and in the indicated other lines the authors talk about hyperphosphatemia. The concept was not introduced in the Intro section and is  not addressed further in the remainder of the review. It appears to be a typo or part of a section that was not further developed. Please emend accordingly 

Author Response

We greatly appreciate the efforts of the reviewers for the time they invested in evaluating our manuscript, and believe that the revised manuscript is significantly improved. We perceived that the reviewers were generally supportive, but did have suggestions to improve the manuscript. We have adapted the manuscript to address each suggestion.

We deleted the 2.4 section, which was a result of typographical error.

Reviewer 2 Report

The manuscript reviews the published investigation on alcohol exposure and levels of Fe, Mg, Cu, and Mn. The review provides a good overlook on the subject.

One of the concerns is the repetition of some section in the manuscript. It is recommended that authors avoid the repetition.

Following mistakes should

Line 135: The words “CTD, ALT, GGT and MCV” should be written in full (wherever they come first).

Line 263: Correct the sentence “where thins condition was not observed”

Line 354: put a full-stop/period in the sentence : non-alcoholic fatty liver disease (NAFLD)”.

Line 355: convert “performer” into “performed”

Line 368-369: Correct the sentence “Similarly, the study by Ordak et al. found decreased copper in blood concentration among 20.4% or 368 the examined patients with alcohol abuse history [53].”

Line 372: Write “Hepatitis C virus for HCV and put HCV in small bracket (HCV).

Line 467: Change “to” to ‘so”

Author Response

We greatly appreciate the efforts of the reviewers for the time they invested in evaluating our manuscript, and believe that the revised manuscript is significantly improved. We perceived that the reviewers were generally supportive, but did have suggestions to improve the manuscript. We have adapted the manuscript to address each suggestion.

We deleted the 2.4 section, which was a result of typographical error.

Following mistakes should

Line 135: The words “CTD, ALT, GGT and MCV” should be written in full - Done

Line 263: Correct the sentence “where thins condition was not observed” - Corrected

Line 354: put a full-stop/period in the sentence : non-alcoholic fatty liver disease (NAFLD)”. - Done

Line 355: convert “performer” into “performed” - Done

Line 368-369: Correct the sentence “Similarly, the study by Ordak et al. found decreased copper in blood concentration among 20.4% or 368 the examined patients with alcohol abuse history [53].” - Corrected

Line 372: Write “Hepatitis C virus for HCV and put HCV in small bracket (HCV). - Done

Line 467: Change “to” to ‘so” - Done